# CLAP: A Cross-Layer Analytic Platform for the Correlation of Cyber and Physical Security Events Affecting Water Critical Infrastructures

**Gustavo Gonzalez-Granadillo** [1,*]**, Rodrigo Diaz** [1]**, Juan Caubet** [2] **and Ignasi Garcia-Milà** [3]

1. Atos Research & Innovation, Cybersecurity Unit, 28037 Madrid, Spain; rodrigo.diaz@atos.net
2. Eurecat, Centre Tecnològic de Catalunya, IT & OT Security Unit, 08005 Barcelona, Spain; juan.caubet@eurecat.org
3. Worldsensing, Barcelona, 08014 Barcelona, Spain; igarciamila@worldsensing.com
* Correspondence: gustavo.gonzalez@atos.net

**Abstract:** Water CIs are exposed to a wide number of IT challenges that go from the cooperation and alignment between physical and cyber security teams to the proliferation of new vulnerabilities and complex cyber-attacks with potential disastrous consequences. Although novel and powerful solutions are proposed in the literature, most of them lack appropriate mechanisms to detect cyber and physical attacks in real time. We propose a Cross-Layer Analytic Platform (denoted as CLAP) developed for the correlation of Cyber and Physical security events affecting water CIs. CLAP aims to improve the detection of complex attack scenarios in real time based on the correlation of cyber and physical security events. The platform assigns appropriate severity values to each correlated alarm that will guide security analysts in the decision-making process of prioritizing mitigation actions. A series of passive and active attack scenarios against the target infrastructure are presented at the end of the paper to show the mechanisms used for the detection and correlation of cyber–physical security events. Results show promising benefits in the improvement of response accuracy, false rates reduction and real-time detection of complex attacks based on cross-correlation rules.

**Keywords:** data analytic platform; cyber–physical security; cross-correlation; water critical infrastructures

## 1. Introduction

Protecting Critical Infrastructures (CIs) is of vital importance and should not only depend on a list of security policies and rules. Human lives depend on the continuous delivery of services such as water, electricity, transportation, etc. Interruptions in any of these essential systems, even if only for a short period of time, could be translated into severe consequences to citizens and organizations.

Unlike traditional IT infrastructures, CIs include two types of technologies that do not always coexist jointly: Information Technology (IT) and Operational Technology (OT). The former includes all forms of technologies used to create, store, share and transmit information (e.g., data, voice, video, etc.), whereas the latter consists of hardware and software systems that monitor and control physical equipment and processes to manage critical infrastructures (e.g., water, gas, energy, transportation, manufacturing, etc.). Both IT and OT have been conceived and managed as two separate organizational entities. The disconnect between these two units has generated unreliable outputs over the past few decades, and the significant benefits of convergence, such as insight into security risks and enhanced performance, demand attention across the technology landscape [1,2].

Despite the advances in the area, Critical Infrastructures are prone to a variety of cyber and/or physical security threats. This is due to their heterogeneous nature, their reliance on private and sensitive data, and their large-scale deployment. As such, intentional or accidental exposures of these systems may result into devastating consequences, making

it necessary to implement novel and robust security measures [3]. As most critical infrastructures, the water domain relies on industrial protocols (e.g., Modbus, OPC, Powerlink, DNP3) in their communications, with a myriad of security limitations, among which we can highlight the following [4–9]:

- Lack of authentication mechanism, making it possible for an attacker to enter the system by creating a packet with a valid address, a function code and any associated data.
- Absence of encryption used in the communication messages, making it possible for an attacker to sniff all communications between masters and slaves.
- No Broadcast suppression, making it possible for an attacker to create flooding conditions in all network addresses.
- No checksum, making it possible for an attacker to spoof packets.

In addition, the water sector is exposed to a wide number of IT challenges that go from the cooperation and alignment between physical and cyber security teams to the proliferation of new vulnerabilities and complex cyber-attacks with potential disastrous consequences, which results into a strong demand of cross-knowledge activities involving awareness and training of cyber–physical security related aspects in the water sector [10–12]. In this order of ideas, the water domain does not have specific cybersecurity plans to address unique risks or particular conditions, and as such, the following gaps have been detected [13–15]:

- Huge gap between IT professionals and end users that makes is difficult to trace any systematic training program;
- Discrepancies on the National cybersecurity strategies among EU member States;
- Lack of systematic cooperation with non-governmental entities and public-private partnerships;
- Need for common standards, semantics, and processes implemented in inter-operable solutions;
- Shortage of qualified technical personnel;
- Lack of awareness in cybersecurity aspects;
- Poor bilateral and multilateral collaborations;
- Lack of trust among organizations.

Considering the above-mentioned limitations, challenges and gaps of the water sector, it is imperative the development of solutions to improve cyber and physical security mechanisms in the area. We present in this paper a Cross-Layer Analytic Platform (hereinafter denoted as CLAP), developed for the correlation of Cyber and Physical security events affecting water CIs. CLAP is used to simulate attacks against a victim machine emulating a water CI with PLCs, and security tools (e.g., intrusion detection systems, SIEMs, access control mechanisms, network traffic analyzers, and tools to detect anomalous events in real time).

The contributions of this article are summarized as follows:

- A Data Analytic platform that integrates various tools for the detection of cyber and physical attacks in critical infrastructures;
- The combination of cross-layer data sources to improve detection mechanisms;
- A process that correlates relevant security data coming from cyber and physical sensors through specific detection and monitoring systems;
- An automated process that maps correlated security events with standardized threat categories and/or attack patterns;
- The deployment of the platform over multiple attack scenarios.

The ultimate goal of the platform is to improve the detection of complex attack scenarios in real time based on the correlation of cyber and physical security events affecting critical infrastructures, as well as to assign appropriate severity values to each correlated alarm that will guide security analysts in the decision-making process to prioritize their mitigation actions.

The remainder of this paper is structured as follows: Section 2 presents related work regarding cyber and physical security platforms. Section 3 describes the architecture of our

proposed CLAP solution. Section 4 details the correlation process used to assign severity levels to the correlated events as well as the sharing mechanisms used by the proposed platform. Section 5 details the platform used for testing and validation of our proposed solution. Section 6 illustrates the applicability of our approach with a variety of use case scenarios. Section 7 discusses about the advantages of cross-correlation in current data security platforms. Finally, conclusions and perspective for future work are presented in Section 8.

## 2. Related Works

The detection and mitigation of physical security events in industrial environments, as well as the detection, analysis and control of cybersecurity incidents in information systems have been widely studied as two separate and unrelated issues. However, a small amount of research has been conducted to the detection of attacks against critical infrastructures based on the correlation of cyber and physical indicators. In addition, as Cyber–Physical Systems (CPS) involve a variety of interconnected elements (closely related to the Internet of Things), the integration of these elements within CPS leads to a new dimension namely the Internet of Cyber–Physical Things (IoTCPT) for which new vulnerabilities, threats and attacks have arisen [3,16]. Table 1 presents a summary of the most recent scientific research in the area and briefly describes their main advantages and shortcomings.

**Table 1.** Cyber–Physical approaches in Critical Infrastructure Systems.

| Approach | Advantages | Shortcomings |
|---|---|---|
| Security platforms to detect cyber-attacks in cyber–physical systems [17,18] | High accuracy on the attack detection and a great effectiveness of the proposed simulation scheme. | Platforms are constrained to the detection of a limited number of known attacks and no information is provided for the case of zero-days or unknown attacks. |
| Non-stationary watermark scheme to detect cyber and physical attacks against critical systems [19–21] | Detects adversaries using non-parametric methods. | It is not equally effective against adversaries using parametric identification methods. |
| Stress-testing platform for cyber–physical water infrastructure [22,23] | Modeling a variety of cyber–physical attacks affecting the normal operation of SCADA elements (e.g., sensors, actuators, PLCs), assessing their impact in what-if exploration scenarios while retaining enough fidelity in the representation of interacting processes and information flow in the cyber layer. | Analysis is performed in an offline basis (i.e., obtaining evidence after the incidents have occurred), which is quite useful to protect critical infrastructures from cyber–physical attacks at an strategic and tactical level, but not to at an operational level that requires an immediate reaction mechanism. |
| Traditional anomaly detection techniques[24–28] combined with density-based and parametric algorithms [29] | The use of multi-stage techniques that isolates both local and global anomalies will generally yield better anomaly detection results. The proposed approach outperforms the density-based techniques and has comparable results to the parametric algorithms. | The proposed technique although provide promising results on anomaly detection it needs to test it using high-dimensional datasets. In addition, this approach focuses only on machine learning techniques, leaving aside complimentary options such as rule-based techniques, which are very useful in the detection of some cyber–physical attacks. |
| Cyber–physical attack detection and correlation system [30] | The approach detects various cyber–physical attacks (e.g., unknown, known, simultaneous attacks) affecting current and future manufacturing systems; reduces the number of alarms, and reduces the false rates. | Some attack types (e.g., intellectual property theft) are not compatible with the proposed cyber–physical attack detection and correlation method. In addition, ML algorithms have been selected based on previous work and need to be compared with other approaches. |

**Table 1.** *Cont.*

| Approach | Advantages | Shortcomings |
|---|---|---|
| Multi-layer cyber–physical system (CPS)-based management framework [31] | Enables supervision, subsystem interoperability, and integrated optimization of urban water cycle, offering significant improvements in resource sustainability and environmental protection in the overall water cycle. | The proposed approach needs deeper research about interconnection of its various layers and interfacing with real systems. |
| Cyber–physical threat detection platform based on a Building Threat Monitoring System [32] | The platform simulates different types of attacks against hospitals (events, incidents, and alerts are transmitted by sensors from healthcare organizations in real time) using various prototypes to assure the security of the personal and patients from various hospitals. | Besides healthcare organizations, the platform is not implemented or tested anywhere else. |
| Risk-based methodology for identifying and assessing IoT-enabled attack paths against critical cyber–physical systems [33,34] | The methodology models cyber and physical interactions using an attack tree topology and relies on well-known standards e.g., CVE, CVSS to identify and assess hidden and/or underestimated cyber–physical attack paths. It reduces false positives and assist decision makers in mitigating multi-hop attack paths. | The main target is on IoT attacks, leaving aside other type of cyber–physical attacks. In addition, it identifies known attacks, but it is not able to detect attacks for which information is not already available in public databases such as CVE, CVSS, and although attack paths models are useful to define mitigation strategies, they can only be defined statically (real-time mitigation is not proposed). |
| Autonomous mitigation of cyber risks in the Cyber–Physical Systems [35] | Provides an Autonomous Response Controller (ARC) that uses hierarchical risk correlation to model attacker paths and measures financial risks faced by the cyber–physical system assets. Results show ARC provides a response faster than intrusion response systems. | The proposed risk assessment and ARC needs to be evaluated on a large scale and applied in other networks (e.g., 5G and could computing systems). In addition, the approach is comparable to an intrusion detection and prevention system, but not to a complete platform composed of multiple sensors and devices that offer correlation, visualization and storage capabilities. |

Although related works search for solutions to appropriate detect cyber–physical attacks in critical infrastructures, most of them perform their analysis in an offline basis (i.e., obtaining evidence long after the incidents have occurred) and rely on either signature-based or anomaly-based detection. CLAP works with both detection mechanisms to correlate in real time (or near-real time) physical and cyber events from multiple sources and generate correlated security alarms with detailed and useful information of the detected attacks, their severity and involved elements. The main shortcoming of our proposed platform, as any real-time detection mechanism using ML and rule-based approaches is the fact that tools must be manually configured and adapted to the target infrastructure. Some tools composing CLAP are not plug-and-play and therefore require a collaborative work between tool owners and end users to work appropriately.

## 3. CLAP: Cross-Layer Analytic Platform

The Cross-Layer Analytic Platform (CLAP) is a simulation environment developed for the correlation of cyber and physical security events affecting Water Critical Infrastructures. CLAP is composed of three main modules: (i) a core module represented by a Cross-Layer SIEM and a Real-Time Anomaly Detector, responsible for the detection and correlation of cyber and physical threat data; (ii) a physical detection module composed of a jammer

detector and a toolbox of technologies for the physical threat protection; and (iii) a cyber detection module composed of a cyber threat sharing service and a toolbox of technologies for security IT and SCADA (as depicted in Figure 1).

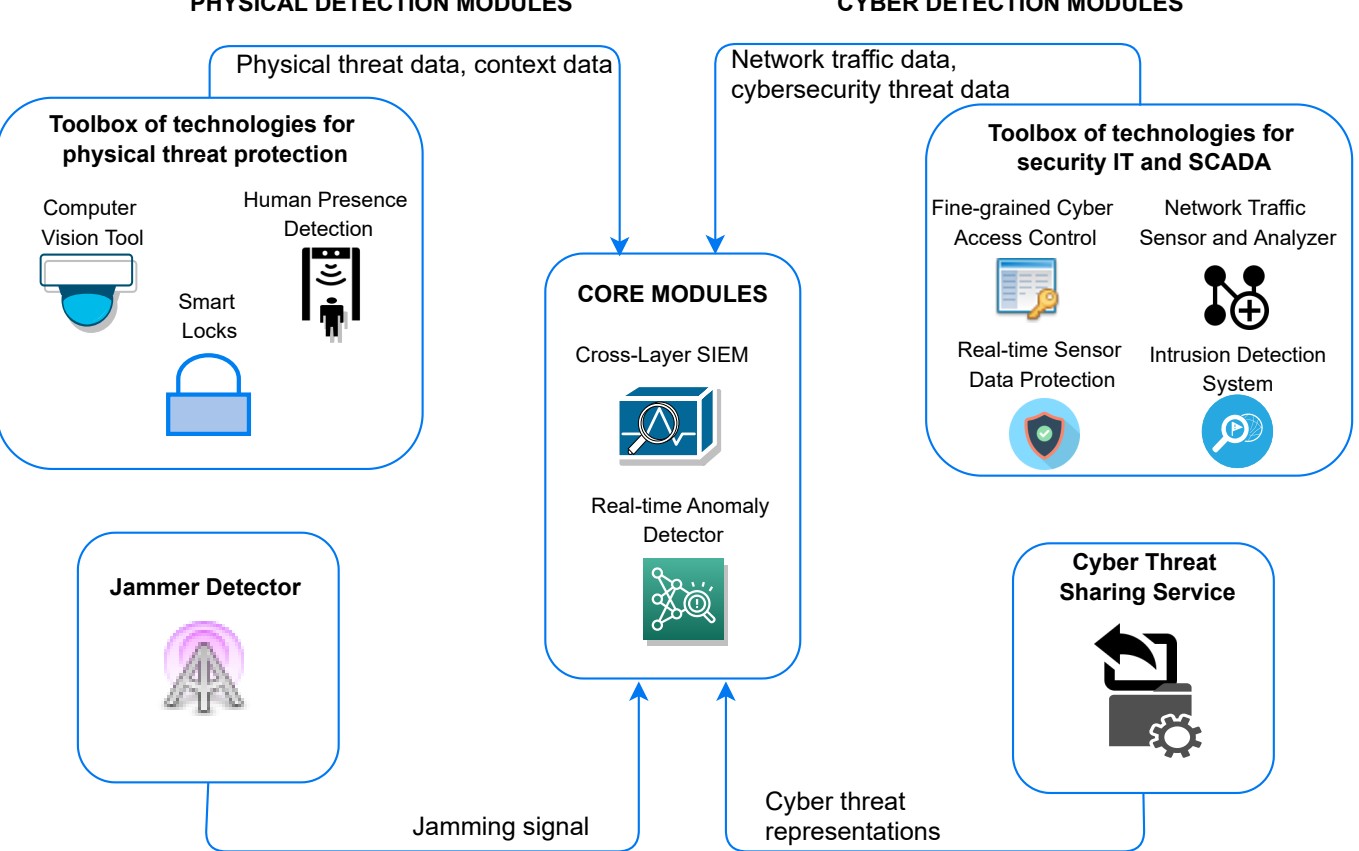

**Figure 1.** CLAP Architecture.

The remainder of this section details each module of our Cross-Layer Analytic Platform.

*3.1. Core Module*

This module aims to detect unknown anomalies with automatic learning abilities for real-time anomaly detection of combined threats and attacks. It is composed of two main tools: (i) The Cross-Layer Security Information and Event Management (denoted by XL-SIEM (https://booklet.atosresearch.eu/xl-siem, accessed on 5 June 2021)); and (ii) the Real-Time Anomaly Detector tool (denoted by RTAD (https://euhubs4data.eu/services/eurecat-artificial-intelligence-and-data-analysis/, accessed on 5 June 2021)).

3.1.1. XL-SIEM

This tool is an enhanced SIEM with added high-performance correlation and able to raise alarms from a business perspective by considering different events collected at different layers [36]. The XL-SIEM is composed of a set of distributed agents, responsible for the event collection, normalization and transfer of data; an engine, responsible for the filtering, aggregation, and correlation of the events collected by the agents, as well as the generation of alarms; a database, responsible for the data storage; and a dashboard, responsible for the data visualization in the graphical interface.

### 3.1.2. RTAD

It addresses the construction of a system to detect unknown anomalies (not based on heuristic tools, lists, or threats already detected) using different sources of information, with automatic learning abilities, and with the supervision of a specialist to validate complex threats to be included in the knowledge base of the system [37]. Context analysis will include interdependencies with other infrastructures (ICT networks, power supply, etc.), social networks, or information that may directly affect its security and resilience.

Both XL-SIEM and RTAD process data coming from the physical and cyber sensors. RTAD maps the correlated events with a standard knowledge base of adversary tactics and techniques to derive the presence of cyber and/or physical attacks (More details are provided in Section 4.3.3).

### 3.2. Physical Detection Module

This module is composed of a variety of tools for protecting the system against physical threats. It is composed of two main elements: (i) Jammer Detector (denoted by JDet (https://www.worldsensing.com/news/unprecedented-security-solution-cipsec/, accessed on 5 June 2021)); and (ii) Toolbox of technologies for physical threat protection [38].

### 3.2.1. Jammer Detector

This tool is composed of a jamming detection sensor with monitoring software that analyzes the radiofrequency spectrum using Software Defined Radio (SDR) techniques and software to detect and inform about wireless jamming attacks. The outcome of this component is a set of logs describing the detected attacks. Furthermore, a friendly visualization interface allows the visualization of attacks in real time. The tool ensures the proper status and availability of the wireless channels, free from physical denial of service attacks.

### 3.2.2. Toolbox of Technologies for Physical Threat Protection

This toolbox is composed of three tools including (i) Computer Vision Tools (denoted by CVT), for automated surveying of the large-area of the water utility; (ii) Smart Locks (https://www.youtube.com/watch?v=UFarfWgGP0w, accessed on 5 June 2021) for the physical access control management; (iii) Human Presence Detector (denoted by HPD), to process and analyze the changes on the WiFi spectrum to detect the movement of intruders in an area which has WiFi coverage.

### 3.3. Cyber Detection Module

This module is composed of a variety of tools for the real-time detection of cyber threats. It is composed of two main elements: (i) Cyber Threat Sharing System (denoted by CTSS (https://stop-it-project.eu/download/cyber-threat-sharing-system/, accessed on 5 June 2021)); and (ii) Toolbox of technologies for securing IT and SCADA [39].

### 3.3.1. Cyber Threat Sharing Service

This tool is in charge of collecting information of threats and attacks from several sources (both internal and external), and providing preventive and mitigation actions to be taken according to the existing systems in the critical infrastructure. CTSS provides automated information sharing for cybersecurity situational awareness, real-time network defense and sophisticated threat analysis.

The tool uses widely accepted standards to describe and share information about cyber threats e.g., the Trusted Automated exchange of Indicator Information (TAXII (https://oasis-open.github.io/cti-documentation/taxii/intro.html, accessed on 5 June 2021)) and the Structured Threat Information expression (STIX (https://oasis-open.github.io/cti-documentation/stix/intro, accessed on 5 June 2021)). They define a set of services and message exchanges that when implemented, enable sharing of actionable cyber threat information across organizational, product line and service boundaries. The main outcome

of the systems will be a graphical interface to analyze information about threats and attacks, observing their relationships, and being able to filter the information according to the existing systems in the critical infrastructure.

3.3.2. Toolbox of Technologies for Cybersecurity Threat Data

This toolbox is composed of technologies for SCADA and IT systems to monitor and protect their integrity both against intentional attacks and/or malfunctions. They include (i) a Fine-grained Cyber Access Control (denoted by FCAC) for the access control management of the cyber and physical entities; (ii) a Real-time Sensor Data Protection (denoted by RSDP) which applies blockchain schemes to protect the integrity of all the data generated during a CI operation (logs, sensor data, etc.); (iii) an Intrusion Detection System (denoted by IDS) to capture system logs about security incidents; and (iv) a Network Traffic Sensor and Analyzer (denoted by NTSA), a machine learning-based tool that monitors network traffic taking place in the managed infrastructure and performs a NetFlow analysis of the network traffic data to accurately detect, in real-time, anomalies that might represent attacks to the infrastructure [40,41].

## 4. CLAP Correlation and Information Sharing Services

This section describes CLAP correlation and information sharing capabilities that allows the processing of data from multiple sources and layers, assign a risk level based on the event severity and share the generated alarms with other systems and communities.

*4.1. Correlation and Cross-Correlation Process*

Previous to the correlation of the collected events, there is a phase of Pre-processing and Policy Filtering, in which the platform verifies if the user has specified some conditions to filter the incoming events before they arrive to the correlation engine (e.g., source/destination IP, port, time/date range, type of event, or the SIEM agent where the event is collected). These filtering conditions help in reducing the volume of events arriving to the correlation engine (and consequently improving its performance), as well as on dividing the processing in different correlation processes (e.g., to have separated correlation by each client organization in the case of multi-tenant deployment). In addition, this pre-processing is important to determine the type of data included in each field of the normalized incoming event (e.g., string, float, integer, double, etc.) since it is relevant for providing a high-performance correlation.

Cross-layer correlation includes security events from multiple layers (e.g., application, transport, network layers). They are generated from different sensors, using different formats and generally referring to different situations with a common parameter (e.g., IP address, protocol, port number, etc.). The Correlation Engine is the core of the XL-SIEM suite and uses Event Processing Language (http://esper.espertech.com/release-5.4.0/esper-reference/html/event_patterns.html, accessed on 5 June 2021) (EPL), which allows a flexible and complex definition of the correlation rules. It is a SQL-like language that includes for example the detection of patterns, the definition of data windows or the aggregation and filtering of incoming events into more complex events.

Since the architecture of the monitoring engine is designed to have its processes running in a distributed way in an Apache Storm cluster, it is possible to define different correlation processes called "correlation bolts". It is then possible to have a different set of rules (different security directives) and a different data schema (different type of incoming events) for each correlation bolt working in separate hosts.

The proposed Cross-Layer Analytic Platform follows a four-step correlation process: (i) define the security policy; (ii) Define correlation bolts; (iii) define statements; and (iv) define directives.

### 4.1.1. Definition of Security Policies

Security policies typically address the constrains on functions and flow among the systems and how users must access them. CLAP uses these policies to filter the events associated with one or more data source agents. It is also possible to define sources or destination of the events, either by filtering the IP addresses ports, or event type. Security policies are used to define the events to be considered for correlation and to define the correlation bolts in the final step.

The definition of security policies must define the source and destination IP addresses or range of addresses affected by the policy, the source and destination port numbers, the type of detected events and the sensor, tool or equipment (agent) responsible for the detection. A regular security policy uses the following structure: (IP_Src, IP_Dst, Port_Src, Port_Dst, Event_Type, Agent).

Every matching policy has an associated action to be executed. The format used to define a policy consequence in CLAP consists of four parameters: (Action, SIEM, Logger, Forwarding). The Action parameter defines the automatic reactions to be executed after the generation of an alert; the SIEM parameter defines actions to be executed by the SIEM for a particular event (e.g., Set Event Priority, Logical Correlation, Cross-correlation); the logger parameter allows the generation of logs; and the Forwarding parameter allows forwarding events to another agent or entity.

Once the policy and consequences are defined, we build the correlation rules. This is a three-fold process that requires (i) the definition of statements; (ii) the definition of directives; and (iii) the definition of correlation bolts.

### 4.1.2. Definition of Statements

A statement specifies the types of events and creates a data structure where to store them. Internally the platform creates a table in a database for these events, which are used by the directives for its correlation and further identification of anomalies. A statement is composed of a unique name and a value that indicates the variables and the sources from which the information will be evaluated. Statements do not fire alerts, but the generated output is used by other services. Table 2 provides Examples of EPL statements.

**Table 2.** Examples of EPL Statements.

| Name | EPL Statement |
|---|---|
| Failed_SSH_ login | insert into Failed_SSH_Login select * from schema_default where (plugin_id in 1001, 2001) and (plugin_sid = 1) |
| jammer_ detector | insert into jammer_detector select * from schema_default where (plugin_id = 100131) |
| abnormal_obs | insert into abnormal_obs select * from schema_default where (plugin_id = 100001) and (plugin_sid = 3) |
| unauthorized_ write_PLC | insert into unauthorized_write_PLC select * from schema_default where (plugin_id = 1001) and (plugin_sid = 1111007) |

It is worth noting that EPL statements require the specification of two variables: (i) plugin_id corresponding to the data sources (e.g., IDS, JDet, NTSA, FCAC); and (ii) plugin_sid corresponding to the type of events we want to consider in the correlation. We can specify one or more events for a particular plugin by indicating its plugin_sid. If we want to consider all events associated with a particular plugin, we need to specify only the plugin_id (as done with the jammer detector statement).

### 4.1.3. Definition of Directives

Directives contain the pattern used in the evaluation of the rule. They fire an alert for every matching pattern. Directives use the statements previously defined to identify

the events that are affected by every rule. A directive is composed of a numeric identifier (ID) that will be used in the correlation bolt definition, a name and an EPL pattern (as depicted in Table 3). Additional fields defined in the Directive panel are the category and subcategory of the alert that will be generated with this directive and the reliability and priority of such directive. These values, together with the ones for the events and the importance of the assets are used to define the severity level of the generated alarm (as detailed in Section 4.2).

**Table 3.** Examples of EPL Directives.

| Name | EPL Statement |
|---|---|
| Failed_SSH_ login | pattern [every-distinct (a.src_ip, 60 s) a = FailedSSHLogin-Event ⟶ [6] b = FailedSSHLoginEvent ((b.src_ip = a.src_ip) and (b.dst_ip = a.dst_ip))] |
| Jammer Detector | pattern [every a = jammer_detector (a.userdata2 = "Attack Running")] |
| Abnormal observation on DST_IP | pattern [every-distinct (a.src_ip, 60 s) a = abnormal_obs] |
| Unauthorized PLC data modification | pattern [every-distinct (a.userdata1,a.plugin_sid, 60 s) a = Unauthorized_write_PLC] |

The EPL directive shows the correlation rule used to fire an alarm if there is a match in the pattern defined within the brackets. The first example indicates that every sequence considering distinctively the IP source and 60 s of timeout will generate an alert, after which the fields are cleared and if an event is received with an IP source that had been previously received, it will be considered to be another alert. For this example, we have defined a failed SSH login as the event to check, if the defined condition matches (i.e., the IP source and destination are the same) and the same event occurs more than 6 times, then a correlated alert will be generated.

### 4.1.4. Definition of Correlation Bolts

After the definition of policies and directives, they need to be linked for an appropriate correlation. To do so the platform uses correlation bolts, which specifies what are the policies to take and what are the directives to consider when looking for anomalies and generate alerts. This step is as important as the previous ones since it activates the correlation rules.

### 4.2. Calculation of Alarm Severity

Each detected event poses a particular risk to the infrastructure. We measure quantitatively the risk severity in a scale from 0 to 10, where 0 refers to the lowest severity and 10 refers to the highest severity. The proposed Cross-layer Data Analytic platform computes a severity score of each correlated alarm based on Equation (1).

$$Severity = \frac{Priority \times Reliability \times Asset_{Value}}{100} \tag{1}$$

The priority parameter represents the importance of the attack if it is realized. It ranges from zero to ten (as the minimum and maximum priority values respectively). A priority value is assigned when the type of event is defined in the monitoring engine for a specific sensor or specific security directives or rules, but it can be modified using a policy to prioritize events in specific situations or conditions.

The reliability parameter indicates the probability for an attack to be successfully executed. It ranges from zero to ten (as the minimum and maximum reliability values respectively). Like priority, this value is assigned based on expert knowledge for a specific

security directive or rule. This value is generally low if the event is detected by only one data source, and it increases if the same event is detected by multiple data sources and become a correlated alarm. The higher the number of data sources, the higher the reliability of the information and thus, the higher the severity level.

The asset value parameter represents a value associated with the importance of the host being attacked in the infrastructure monitored. It ranges from zero to ten (as the minimum and maximum importance asset values respectively).

The computed severity indicates the potential impact associated with a correlated alarm. It is defined to help security analysts in prioritizing their mitigation actions. Correlated cross-layer events imply higher severity than correlated singer-layer events, as the former provides a wider vision of the incident with a higher reliability that the latter.

### 4.3. Information Sharing and Exporting

Our proposed platform collects data from multiple sources and normalize this input data into a common format for further analysis and correlation. The outcome of this tool are events and correlated alarms indicating errors, attacks and any kind of malicious incidents on the target system. Such alarms are shared with other tools and/or components using a standard format (e.g., STIX, JSON) and can be exported as a PDF report.

#### 4.3.1. Alarm Sharing

CLAP supports sending alarms (in JSON or STIX formats) via RabbitMQ (https://www.rabbitmq.com/, accessed on 5 June 2021), a widely used open-source message broker that supports multiple messaging protocols. It also supports sending alarms via the Data Distribution Service (DDS), commonly used for real-time systems. Events generated by the XL-SIEM agent from the data collected by the sensors are sent in JSON format to a RabbitMQ server.

RabbitMQ is used to create queues and provide secure communications between some of the CLAP components. RabbitMQ works as a passive component; therefore, other tools will connect to RabbitMQ to exchange information using the queues. RabbitMQ needs to be configured to provide secure communications and to create the elements (queues and exchanges) needed for the communications among CLAP components.

CLAP gives the possibility to enable external and internal authentication mechanisms for the clients connecting to the RabbitMQ. When having just the external mechanism enabled, clients will not be able to connect without a signed certificate. Once authenticated, clients can check the queue activity by accessing to the RabbitMQ web administration page.

The proposed solution sends alarms generated by the XL-SIEM engine to a particular queue, which will be accessed by the Real-time anomaly detector tool for further analysis and visualization.

#### 4.3.2. Exporting Outputs

CLAP generates a PDF report containing the list of generated alarms within a predefined period of time, the top 10 detected attacks, most used ports, the hosts involved in the security alarms, the number of occurrences of each alarm, and the top 15 alarms based on their risk level. Automatic actions can be specified to deal with the platform output e.g., a PDF report can be automatically generated with a predefined periodicity, it can be sent by email or displayed in the platform dashboard.

It is also possible to generate a report of the detected events either in CSV or PDF format. The report allows the display of up to 5000 events and includes several data features such as the type of event, the timestamp, the sensor responsible for the detection, the IP and port source and destination, as well as the severity level associated with the event.

#### 4.3.3. Security Alarms and CAPEC Mapping

Sharing events and/or alarms with other organizations, communities and groups encourages collaboration among researchers and help improving the detection of complex

attacks in the domain of interest. CLAP outcomes are mapped with attack patterns from the Common Attack Pattern Enumeration and Classification (CAPEC (https://capec.mitre.org/, accessed on 5 June 2021)), aiming to advance community understanding and enhance defense mechanisms. Examples of the correlated alarms mapped with CAPEC is shown in Table 4.

**Table 4.** Correlated alarms and CAPEC Mapping.

| MITRE | Technique | CAPEC | Description |
|---|---|---|---|
| T1110 | Brute Force | CAPEC-49 | Failed SSH Login |
| T1498 | Network DoS | CAPEC-125 | Jammer Detector |
| T1057 | Process Discovery | CAPEC-573 | Abnormal observation on DST_IP |
| T1078 | valid Accounts | CAPEC-560 | DENIED access from SRC_IP to DST_IP |

## 5. Platform Testing and Validation

For the testing and validation of our proposed platform, we have developed a testbed scenario within the context of the European Project STOP-IT (https://stop-it-project.eu/, accessed on 5 June 2021), composed of an attacker machine, a victim infrastructure and a CLAP instance.

As depicted in Figure 2, the attacker initiates passive and active attacks to identify the active devices on the network and exploit some vulnerabilities (1). Once the IDS detects the intrusion, it generates events to feed the XL-SIEM (2). As soon as radiofrequency anomalies are detected by the Jamming Detector (3), it sends an event to the XL-SIEM (4). In parallel, the NTSA is listening to the network to check for anomalies on the traffic (5). Once an anomaly is detected, it sends an event to the XL-SIEM (6). Events from all sensors are correlated by XL-SIEM and shared with the RTAD for further processing (7). The remainder of this section will describe each of the components from the testbed scenario.

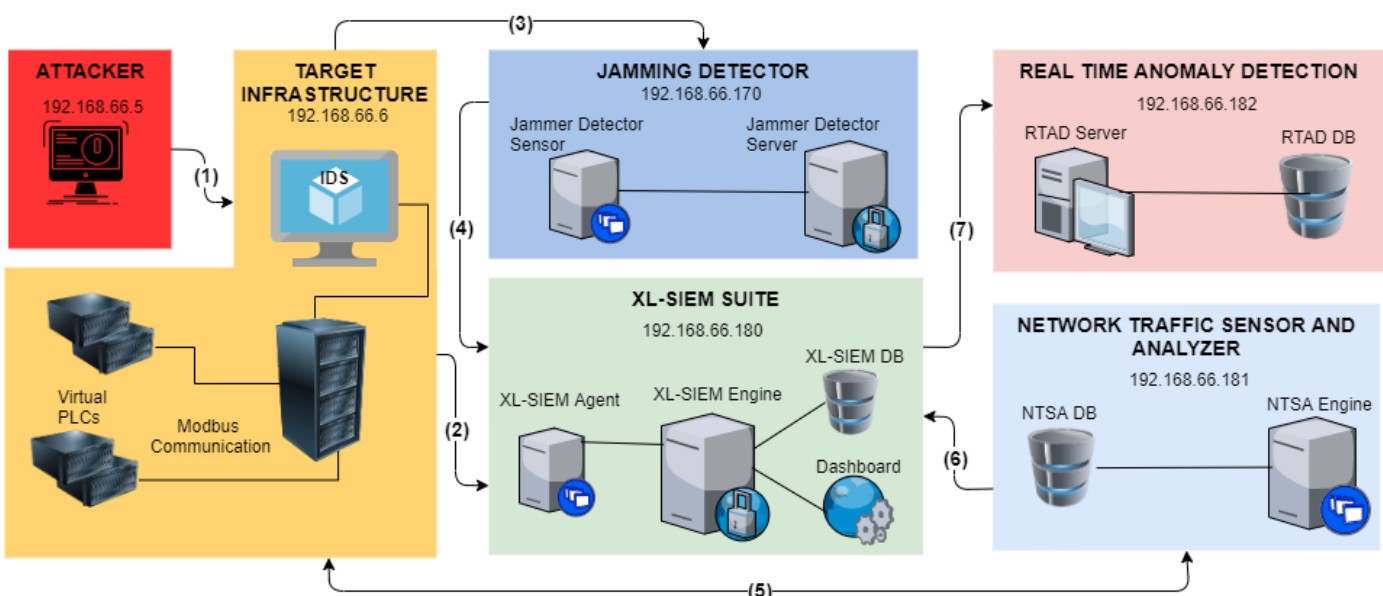

**Figure 2.** Testbed Scenario.

### 5.1. Attacker Machine

It contains a Kali Linux distribution (IP address 192.168.66.5) aiming to execute malicious actions to attack the victim, gain access to the system and being able to read and write values to the database. A typical attack to the SCADA network needs to exploit the SCADA devices and the protocol vulnerabilities. Taking for instance a vulnerable PLC,

a hacker can use the Internet to access the web console of the device and gain the user privilege through code injection; after that, he/she can forge a Modbus command to force all the slaves offline.

A normal process in this scenario pumps water from the Dam to the reservoirs and tanks up to the citizen houses with a constant temperature within the predefined threshold (in our case 40 °C). The temperature measurement generated by all sensors is stored in a local database. An attacker may try to read and/or modify this information without being noticed.

Several attacks can be performed against the SCADA testbed scenario previously described. We can distinguish two main attack types (i.e., passive and active) against major security services e.g., authentication confidentiality, integrity, availability.

*5.2. Target Infrastructure*

It emulates a water CI composed of a water Dam, pumps, tanks and water treatment labs with sensors that measure the temperature and provides a measurement in several key points from the whole water distribution chain (as depicted in Figure 3). The average temperature is set to 40.0 °C (default value in this test scenario), with minimum and maximum threshold values set to 40.0 °C. The measurement of the sensors is performed every hour by the simulated PLCs and the corresponding data are stored in a local database.

The Water distribution scenario has been developed using Rapid SCADA (https://rapidscada.org/, accessed on 5 June 2021), an open-source industrial automation platform that provides tools for rapid creation of monitoring and control systems. The main objective is to simulate SCADA elements from a water CI and perform abnormal/malicious actions against the data generated by the Programmable Logic Controllers (PLCs) that is stored in the local databases. These actions should be detected by security devices, and alarms must be generated accordingly to warn the SCADA operators about the presence of a potential threat or attack.

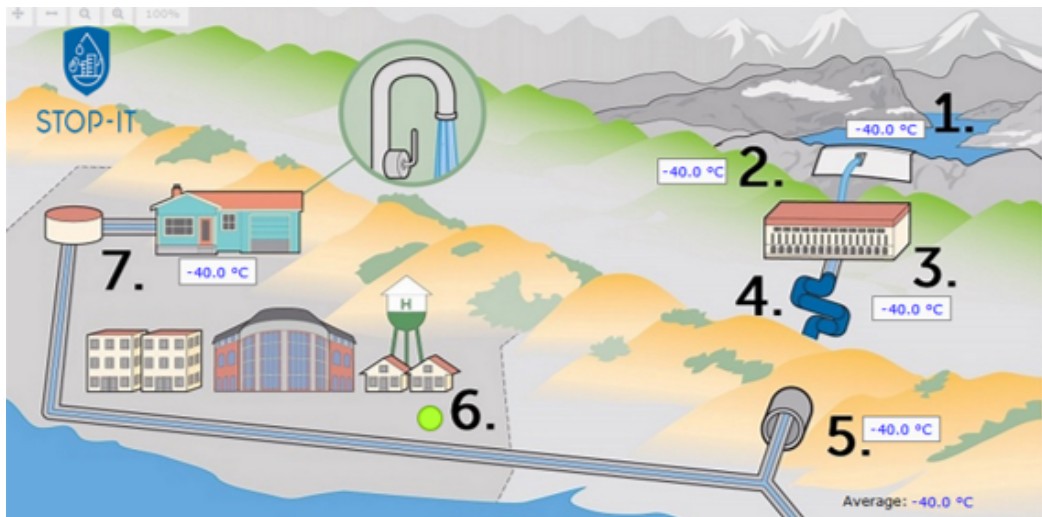

**Figure 3.** Water Distribution Scenario.

The water CI has Ubuntu as the operating system with an IP address 192.168.66.6 and contains a working SCADA system with simulated PLCs and a Modbus (http://www.modbus.org/, accessed on 5 June 2021) Slave. Modbus is a SCADA oriented protocol used for transmitting information over serial lines among electronic devices. Unlike conventional IT networks that use as protocols HTTP, FTP, and SNMP, industrial networks use proprietary protocols such as Modbus, Powerlink and DNP3 in their communications. The device requesting information is known as the Modbus Master, whereas the one that provides the information is known as the Modbus Slave [42].

*5.3. CLAP Instance*

It contains an instance of the XL-SIEM and the RTAD tools along with a variety of data sources for the detection and analysis of cyber and physical security events. The XL-SIEM agent receives events coming from the data sources deployed in the target infrastructure and translates this information into a common format. Events are correlated and alarms are generated accordingly, indicating the presence of an attack in the system. All events and alarms are stored in an internal database, displayed in the Dashboard, and shared with the RTAD tool (IP address 192.168.66.182) for further analysis. More details about the XL-SIEM can be found in [36].

Due to space constrains, we selected four data sensors (i.e., Suricata, Jammer detector, and Network Traffic Sensor and Analyzer) to feed the CLAP central node with cyber and physical security data. The remainder of this section details each security sensor composing the testbed scenario.

5.3.1. Sensor 1: Suricata

Suricata is an Intrusion Detection System (IDS) installed and launched in the target infrastructure (192.168.66.6) as a security sensor of the CLAP. By the time the testbed was developed, we used Suricata version 4.1.3 with default rules, as well as security rules for attacks against SCADA oriented protocols. The logs produced by the IDS are sent to the XL-SIEM agent for further processing to feed the XL-SIEM engine and generate correlated alarms accordingly. As such, the platform can detect security issues, policy violations, and any kind of malicious/suspicious activities that generate logs in the target infrastructure, based on predefined security rules.

5.3.2. Sensor 2: Jammer Detector (JDet)

The JDet (IP address 192.168.66.170) is a sensor of the CLAP that identifies anomalies on the radiofrequency spectrum where the sensor is located. The signal is received by physical device(s) and is transformed into an alert that is then shared with the XL-SIEM agent. A plugin has been developed to allow communications of the JDet and the CLAP. The jamming events are sent to a specific IP address using a predefined format, which will trigger an alert in CLAP. JDet generates an alert when a jammer generates noise in the area covered by the sensor.

5.3.3. Sensor 3: Network Traffic Sensor and Analyzer (NTSA)

The NTSA (IP address 192.168.66.181) uses unsupervised algorithms to create a model of the normal behavior of the system e.g., by modeling the number of packets transferred during a given period of time, the volume of packets sent/received, the IP sources/destinations used in the communications, the port sources/destinations required for communications, the protocols used, etc., therefore, any traffic data falling outside the model will be considered to be suspicious, and the tool will alert the systems accordingly.

NTSA is also a sensor of the CLAP for cyber security data of the network traffic. The main output of this tool is a list of Warnings and/or Errors that indicate unknown detected IP, or any kind of abnormal observation, as well as detected IP that has not been modeled. More details about the NTSA suite can be found in [40,41].

*5.4. Platform Validation*

The validation methodology focuses on the end-user experience (UX), gained during demo activities on both an individual tool level (e.g., the use of a single tool) and a platform level (e.g., the experience obtained from the use of CLAP). The methodology is based on identifying key parameters (named tool and platform traits) that characterize tool and platform performance, as seen by the end user, and thus allows him/her to evaluate performance based on qualitative questions linked to these traits.

Seven main trait categories are identified: (i) ease of installation; (ii) facilitation of user learning; (iii) data requirements; (iv) support; (v) integrity; (vi) usability; and

(vii) usefulness. All traits are applicable to both tool and platform level, which formulate seven different chapters with questions for the questionnaires. With questionnaires at tool and platform level, the end users can provide validation scores (in a rank from 1.0 to 5.0) for every trait category. These scores are then aggregated and used to assess the quality of an individual tool or of the platform as a whole. The scores can be further used for relevant reflection activities and to provide feedback to the developers to improve the design of specific tools or platform functions. The validation process is summarized as follows:

5.4.1. End-User Validation

Mekorot (https://www.mekorot-int.com/, accessed on 5 June 2021) is a National Water utility located in Israel and one of the end users that tested and validated our proposed platform. As part of the STOP-IT project, Mekorot selected the Sorek Desalination Intake Plant (https://www.water-technology.net/projects/sorek-desalination-plant/, accessed on 5 June 2021) as its demonstration site. The Sorek intake facility connects the Sorek Seawater Desalination Plant to the Mekorot national water system and is located close to the Tel-Aviv metropolis. The facility is constructed of pumping stations (5 Units), $2 \times 75,000$ m$^3$ storage pools, a local network - PLCs and HMI, and a remote connection (MDLC protocol) to Mekorot SCADA. Its primary goal is to overcome the gap between supply (24/7) and demand (according to the domestic consumers' behavior) since the national water system does not include a large storage volume.

The testing and validation phase occurred both physically (at the Sorek Intake Facility) and remotely (offline) using recorded real data. No live data or connection was made to the Mekorot SCADA system out of security and operational policy considerations. Mekorot's command and control system is managed by nine control centers covering the entire land surface of Israel, and the system operates 3000 remote sites by radio, cellular, and satellite links. For this reason, the Sorek intake facility premises and operational data served as a representative environment for the Mekorot SCADA system, whose main interest is to protect its SCADA system from physical intrusions and cyber-attacks and to improve the system's performance due to its wide-ranging dependency SCADA platform.

5.4.2. End-User's Feedback

Based on the defined validation criteria, Table 5 summarizes the main assessment results obtained out of the quality parameters evaluation performed over the proposed platform [43].

**Table 5.** Quality Parameters Evaluation.

| Parameter | 1 | 2 | 3 | 4 | 5 | Comments |
|---|---|---|---|---|---|---|
| Ease of installation | | | | | • | No major issues identified during installation |
| Facilitation of user learning | | | | • | | Appropriate user guidelines to learn and demonstrate tools/platform capabilities |
| Data requirements | | | | | • | Data anonymization is required prior testing |
| Support | | | | | • | Available during installation and deployment |
| Integrity | | | | • | | Some tools are not plug-and-play, and require adaptation to the water utility |
| Usability | | | | • | | Some tools do not have GUIs; however, the platform uses the XL-SIEM as its major GUI |
| Usefulness | | | | • | | The platform limits its usage to operate only in English environments |

As can be seen from the previous table, the overall quality assessment of the proposed platform is good or very good. it is important to note that besides Mekorot, two other water utilities (i.e., Berlin Wasserbetriebe (https://www.bwb.de/de/index.php, accessed on 5 June 2021), Oslo Water and Sewerage Works (https://www.oslo.kommune.no/politics-and-administration/statistics/environment-status/water-and-waterways/#gref, accessed

on 5 June 2021)) have participated in the platform testing and validation by reflecting how useful the proposed solution is in their operational contexts in terms of (i) accuracy improvement (higher detection of physical/cyber-attacks and incidents); (ii) latency reduction; (iii) response time; (iv) preparedness improvement; (v) reduction of human exposure; and (vi) cost effectiveness. The end users, as tokens of expert judgment, can project and estimate the expected impact the platform will have to their line of work, as well as to the level of protection that can be achieved with the framework. This reflection on projected impact creates a surrogate validation scheme, based on expert end-user judgment, to measure the impact of the platform in the end-user domains.

The following aspects have been considered during testing and validation of the proposed solution:

- Anonymized data are used during the whole process.
- NTSA training dataset must include legitimate connections to develop the machine learning model of the network behavior for each water CI. The testing dataset must contain anomalous connections (e.g., attacks).
- Some of the platform tools (e.g., JDet, NTSA) can operate using physical deployment, others (e.g., XL-SIEM, RTAD) use a cloud-based deployment and others can use both.
- Devices from the end-user infrastructure can be physical, virtualized, or both.
- End users must define their own security policies and rules to detect attacks in their infrastructures.
- The platform is expected to perform the detection analysis in real time.

## 6. Use Case: Cyber and Physical Attacks against a Water Critical Infrastructure

An unauthorized user (hereinafter referred as attacker) tries to access a database server located in a security room from the target organization. After being denied access a couple of times, the attacker succeeds to enter the room physically. A log is generated by Suricata indicating a possible intrusion. Once in the room, the CLAP obtains information about the actions performed by the attacker. NTSA generates abnormal behavior messages indicating unusual connections from/to IP 192.168.66.6 (the command and control node of the water CI). In parallel, a jamming signal affecting the same IP address is received from the JDet, and a big number of security logs related to the target infrastructure have been generated by Suricata. Events with the same IP source/destination are correlated and security alarms with an associated risk severity are automatically raised by CLAP.

This section details the steps taken by the attacker to perform a series of passive and active attacks against the target critical infrastructure, and the mechanisms used by our developed platform to detect and correlate them.

### 6.1. Network Reconnaissance

After being connected to the target infrastructure, the attacker performs passive actions through PING scans (Internet Control Message Protocol – ICMP packets sent to an IP or a range of IP addresses to see which ones respond and are alive); or Port scans to discover services running on a network or target machine. Once scanned, the attacker can plan an active attack against identified vulnerable services.

The objective of this attack is to identify the hosts composing the network. For this purpose, the attacker, knowing that the network address is 192.168.66.0/24, he/she executes a PING command from the attacker machine running Kali Linux. In just a few seconds, the attacker can retrieve the list of IP address of all active hosts in the LAN (i.e., 192.168.66.1, 192.168.66.2, 192.168.66.5, 192.168.66.6, 192.168.66.7, 192.168.66.180).

Once the attacker has confirmed that there are active hosts in the network, he/she may perform a port scanning using Nmap to identify open ports and services running in each host. After the reconnaissance of the network and hosts, the attacker prepares an active attack to read and/or modify information from any of the tables from the system's database.

Suricata generates logs related to a potential policy violation in DHCP request packets. After the execution of the PING requests, Suricata generated logs entitled "ICMP3 Packet

found" provide an identification of the IP source and destination involved. Although the IDS is unable to classify this log under a predefined category, the priority of this log has a level 3, which requires a deeper analysis from the SCADA administrator. Examples of the Suricata messages received by the XL-SIEM are provided as follows:

- 04/03/2020-08:16:14.515429 [**] [1:4000006:0] ICMP3 Packet found [**] [Classification: (null)] [Priority: 3] ICMP 192.168.66.5:8 $\longrightarrow$ 192.168.66.6:8
- 04/03/2020-08:19:15.046731 [**] [1:2022973:1] ET POLICY Possible Kali Linux hostname in DHCP Request Packet [**] [Classification: Potential Corporate Privacy Violation] [Priority: 1] UDP 192.168.66.5:68 $\longrightarrow$ 192.168.66.6:67

These logs are sent to the CLAP to be processed and displayed in the platform's dashboard with the following format:

**<Signature**; **Date**; **Sensor**; **Source**; **Destination**; **Risk>**

Examples of the security events coming from Suricata and displayed by CLAP are given bellow:

- PING test; 2020-04-03 08:16:14; xlsiem-server; 192.168.66.5:8; 192.168.66.6:8; 0
- ET POLICY Possible Kali Linux hostname in DHCP Request Packet; 2020-04-03 08:19:15; xlsiem-server; 192.168.66.5:68; 192.168.66.6:67; 0

Each detected event has been assigned a severity of cero, meaning that the risk level is very low. All ICMP logs are displayed in the CLAP dashboard as PING test events, indicating the time at which they were detected, the sensor responsible for the detection, the IP source and destinations involved, the asset value and the risk level associated with each event.

*6.2. Reading/Writing PLC data*

The attacker launches the Kali machine (192.168.66.5) and starts the PostgreSQL and Metasploit . To execute an attack against the victim database, the attacker uses a Modbus auxiliary from Metasploit. At this point, the attacker has two possibilities: read or write against the victim database. The attacker decides to read info from the database by specifying the target IP address (e.g., RHOSTS 192.168.66.6), the target port number (e.g., RPORT 502) and the Modbus data address (e.g., DATA_ADDRESS 1). After executing the attack, the attacker can read the values of the registers.

In addition, the attacker succeeds to modify the registers in the database. In this simulation, the attacker has added values to the first seven registers, which originates an abnormal situation with temperature values falling outside the threshold.

The detection of these attacks is performed using rules specifically designed for the Modbus protocol (e.g., read write registers, unauthorized access to the port 502, etc.). Examples of these rules used by our IDS can be found in [44].

As a result, after the execution of the attack to read and write registers in the database, two logs have been generated by Suricata entitled: Modbus TCP – Unauthorized Read Request to a PLC and Modbus TCP – Unauthorized Write Request to a PLC. Examples of the Suricata messages received by the XL-SIEM are provided as follows:

- 04/03/2020-09:32:36.325102 [**] [1:1111123:1] Modbus TCP-Unauthorized Read Request to a PLC [**] [Classification: Potentially Bad Traffic] [Priority: 2] TCP 192.168.66.5: 39621 $\longrightarrow$ 192.168.66.6:502
- 04/03/2020-09:43:54.890149 [**] [1:1111007:1] Modbus TCP-Unauthorized Write Request to a PLC [**] [Classification: Potentially Bad Traffic] [Priority: 1] TCP 192.168.66.5: 44919 $\longrightarrow$ 192.168.66.6:502

Examples of the security events involving PLC communications within the platform are provided bellow:

- Unauthorized Read Request to a PLC; 2020-04-03 09:32:36; xlsiem-server; 192.168.66.5: 39621; 192.168.66.6:502; 5

- Unauthorized Write Request to a PLC; 2020-04-03 09:43:54; xlsiem-server; 192.168.66.5: 44919; 192.168.66.6:502; 5

Each detected event has been assigned a severity of four, meaning that the risk level is medium and should be carefully considered for mitigation.

### 6.3. Anomalous Network Traffic

The attacker performs multiple actions that are considered anomalous and that could be detected by a machine learning algorithm. The attacker uses an IP address (i.e., 192.168.66.5) to perform all the commands that would allow him/her to discover and exploit vulnerabilities in the target infrastructure. However, the Network Traffic Sensor and Analyzer has already built a model of the regular behavior of the water CI traffic and is able to detect abnormal actions (e.g., abnormal connections to IP addresses, abnormal protocols used, abnormal ports opened, etc.). For this purpose, we need to obtain a NetFlow of the network traffic.

The NetFlow information considers timestamp (date at which the flow started) a duration of the flow, the protocol involved (e.g., UDP, TCP), IP addresses and ports (source and destination) , as well as the number of packets, size (in bytes) and the number of flows. The NTSA analyzes all this information and focuses on the IP addresses to determine which of them are considered normal (they belong to the regular communications performed in the network), or abnormal (new IP addresses detected and considered to be malicious). The main parameters evaluated are IP location (internal or external to the network); IP distance (distance between the modeled IP and the new ones); and IP knowledge (known or unknown IPs) as described in [GON19].

The NTSA needs to check in real time the network to compare with the created model if there are abnormal situations. Examples of the NTSA messages received by the XL-SIEM are given as follows:

- Dec 18 13:02:41 cyberagent logger Dec 18 13:02:40 172.16.4.199 [L-ADS] ERROR: {src_ip=192.168.66.5, src_port=0, dst_ip=192.168.66.6, dst_port=0, proto=58, desc= "Abnormal observation"}#015
- Dec 18 13:02:41 cyberagent logger Dec 18 13:02:40 172.16.4.199 [L-ADS] WARNING: {src_ip=172.16.23.77, src_port=0, dst_ip=192.168.66.6, dst_port=0, proto=58, desc= "172.16.23.77 not modeled"}#015

Examples of the security events coming from the NTSA and displayed by CLAP are given bellow:

- L-ADS ERROR: Abnormal observation; 2019-12-18 14:02:41; xlsiem-server; 192.168.66.5; 192.168.66.6; 4
- L-ADS WARNING: IP not modeled; 2019-12-18 14:02:41; xlsiem-server; 172.16.23.77; 192.168.66.6; 3

As can be seen in the previous examples, we received two distinct events from the NTSA. One is an error and refers to an abnormal observation in a specific IP address (which will be considered an abnormal event), and the other is a warning indicating that the NTSA was unable to model the IP, and therefore no analysis was performed. Events received from NTSA are assigned a severity of three, meaning that the risk level is low and should be further analyzed.

### 6.4. Jamming Signal

In case the attacker decides to perform a jamming signal to block the communications of the target infrastructure, the cross-layer data analytic platform can use the Jammer detector (JDet) to identify these events in real time and alert the SCADA administrators.

Jamming events are sent to the XL-SIEM agent using a predefined format, which will trigger an alert in the platform. JDet generates an alert when a jammer generates noise in the area covered by the sensor. An example of the events received by the XL-SIEM from JDet is as follows:

- Dec 18 12:24:04 localhost SDRJD[769]: INFO:sdrjdsyslog:{"user": "prototype", "jnr": 20.364, "event_duration": 82025, "nodeId": "3", "srcIp": "172.16.4.235", "dstIp": "192.168.66.6", "time": "2019-12-18T12:22:40.000", "freq": "2412000000", "type": "Pulsed", "event":"Attack Ended"}

Events coming from the JDet and processed by the XL-SIEM are normalized with the platform format. Examples of the security events coming from the NTSA and displayed by CLAP are given bellow:

- Antijamming - Pulsed; 2019-12-18 13:24:03; xlsiem-server; 172.16.4.235; 192.168.66.6; 5
- Jammer detector; 2019-12-18 13:25:41; xlsiem-server; 162.12.144.202; 192.168.66.6; 5

Similar to the log processed from Suricata, the events coming from the JDet indicate a name (signature), a date where the event was received by the CLAP, the sensor in charge of the detection (in this case the xlsiem-server), the IP source and destination (here ports are not indicated), and the risk level (in this case 5 means a medium level of risk).

*6.5. Preliminary Results*

The implementation and deployment of our proposed platform in water critical infrastructures provide the following improvements [43,45]:

- Event Reduction. Water utilities experienced ta reduction of at least two thirds of the security events collected by their cyber and physical sensors. As a result, security administrators do not feel overwhelmed by the huge amount of data to be analyzed.
- Accuracy Improvement. The detection and analysis capabilities have been greatly improved with the use of a wide variety of sensors feeding the core components of the platform with contextual data regarding cyber and physical events in near-real time. As a result, at least one third of the false events are discarded, making the detection process more accurate, as CLAP correlates security data from multiple sources and multiple layers. More precisely, water utilities experienced higher detection of cyber-attacks/incidents and lower false-positive rates (at least 50% better than the current state). This is particularly valuable on the detection of new and unknown threats in the system infrastructure. In addition, the use of a tool such as XL-SIEM maximizes the activity of the C-level managers in the control center, by presenting the alerts on a user-friendly interface and alert management platform.
- Latency Reduction. Using tools such as the NTSA, we are reducing considerably the effort and time required by a security administrator during the detection and analysis of events (especially for zero-days threats). NTSA spots attacks/incidents at least ten times faster than the current technology used by the water utility. In most cases, effort is reduced to less than half, since the use of machine learning algorithms improves considerably the detection of unknown patters and the analysis of abnormal behavior. Overall, the platform enables C-level managers to react in (near) real-time (i.e., in less than 1 min) to security alerts.
- Response Time Reduction. The platform offers (near) real-time response (i.e., in less than 10 s) for interactive users even in the case of a complex correlation of large data volumes (i.e., tens of Terabytes) and/or high data rates (i.e., one or more Terabytes).
- Human Exposure Reduction. Most of the detection activities require manual intervention for their analysis. This is time consuming, error prone, and demand a great effort from security analysts. Preliminary results show that the developed platform helps reducing effort and time during the detection and analysis of unknown malicious events while at the same time renders the process more accurate. We have experienced an effort reduction of at least 50% compared to the manual process performed without the use of CLAP to analyze the detected events.

In addition to the aforementioned improvements, the situational awareness dashboard provided for visualization purposes allows the highlighting of security data related to the particular events and assigns a risk level based on its severity. This helps not only security administrators, but also C-level managers in understanding the risks and possible

actions to take to keep them within acceptable levels. Furthermore, using a multi-level risk management process, the platform can improve the risk assessment associated with each generated alarm depending on the involved sensor. When two or more sensors refer to the same security event, the risk level is higher compared to the same event detected by a single sensor.

## 7. Discussion

The proposed platform processes events received by multiple sources, and generates security alarms accordingly. It is important to highlight that only with the logs provided by one security sensor (e.g., an IDS), CLAP can correlate events and produce alarms to indicate the presence of a threat or an attack in the monitored system. However, as a cross-layer platform, CLAP can correlate events coming from different sources (e.g., NTSA, JDet, IDS, etc.), which will result into alarms with higher impact values and more reliable values (e.g., low levels of false-positive and -negative rates).

Using the built-in SIEM features, CLAP can correlate events coming from the system logs (IDS installed in the end-user infrastructure) with information from the network traffic (generated by the NTSA tool) and information from other security sensors related to jamming signals (JDet).

Security alarms generated in the platform are shared with the Real-time anomaly detector (RTAD) via RabbitMQ for further analysis. Alarms are also displayed in the platform's dashboard with the following format:
<**Signature**; **Events**; **Risk**; **Duration**; **Source**; **Destination**; **Status**>

Examples of the security alarms generated by CLAP are given bellow:

- Policy violation; 3; 3; 0 secs; 192.168.66.5:39058; 192.168.66.6:http; open
- Jammer detector; 2; 5; 0 secs; 172.16.4.235:ANY; 192.168.66.6:ANY; open
- L-ADS ERROR: Abnormal observation on DST_IP; 2; 4; 0 secs; 192.168.66.5:ANY; 192:168:66:6:ANY; open
- Unauthorized PLC data modification; 2; 5; 0 secs; 192.168.66.5:32907; 192.168.66.6:asa-appl-proto; open

In the previous examples we have several alarms that correlates 2 or more events affecting the same IP source and/or destination within a given period of time. In each alarm there is an indication of the incident (signature), the number of correlated events, the severity level associated with the alarm, the duration of the events in seconds (from the first to the last detected event), the IP source and destination, port source and destination, as well as protocols involved, and the status of the alarm (open/close).

It is important to note that the strength of the proposed solution lies in its correlation engine. In this particular example, a policy violation against IP address 192.168.66.6 has been detected by an IDS, the NTSA detects an error on the same IP, indicating an abnormal behavior of this resource and the JDet identifies a jamming signal where the target IP is involved.The platform correlates all these alarms and automatically generate a cross-correlated alarm with a higher severity level.

Considering the information from the individual alarms, their severity ranges from 3 (low) to 6 (medium). They have been generated based on events simultaneously detected by three distinct data sources (i.e., Suricata, NTSA, and JDet), making it possible to correlate them based on the destination IP address. As a result, the following cross-correlated alarm has been generated.
Multiple attacks against IP_DST; 5; 8; 60 secs; 192.168.66.5:ANY; 192.168.66.6:ANY; open

It is important to highlight that the previous alarm has a high severity level (equivalent to 8), which places this alarm in the top of the priorities to be treated by the security analysts. Using cross-correlation rules, we reduce the amount of information (keeping only the most valuable data to the security administrator), which in turns improves the analysis and decision-making process. In addition, more accurate responses (with higher confidence) can be obtained in real time (or near real time), generally within a few seconds. Furthermore, false rates are considerably reduced, as different data sources point to the same incident

and the probability of generating false alarms is very low. As a result, the detection of complex attacks is greatly improved by adding events from a variety of data sources.

## 8. Conclusions

In this paper, we presented CLAP, a Cross-Layer Analytic Platform developed for the correlation of Cyber and Physical security events affecting water CIs. The platform is used to detect cyber and physical attacks against a target infrastructure emulating a water CI with PLCs, and a variety of security tools (e.g., IDS, SIEM, access control mechanisms, network traffic analyzers). The platform is composed of (i) a central node including a SIEM tool and a Real-time anomaly detector tool that receive cyber and physical data from a variety of security sensors; (ii) a physical detection module, including a jammer detector tool and a toolbox for physical threat protection; and (iii) a cyber detection module, including a cyber threat sharing system and a toolbox for security IT and SCADA.

CLAP processes logs and events coming from a variety of sensors and correlate them based on their IP addresses, port numbers, protocols or a combination of them. The correlated security alarms are automatically generated by the platform and shared with a Real-time anomaly detector (RTAD) tool for further analysis. The developed platform deploys cross-correlation rules to group alarms affecting the same IP source or destination and provides a severity level based on multiple parameters (e.g., priority, reliability, asset value). The severity resulting from a single event is generally lower than the one assigned to a security alarm. Cross-correlated alarms provide higher severity values than those assigned to individual alarms. As a result, it is possible to improve detection by reducing the number of false rates and keeping only the most valuable and accurate date to the security administrators.

Future work will focus on adding other sensors to the platform and analyze its performance in terms of correlation, scalability, speed and accuracy on the detection of complex attack scenarios affecting both cyber and physical elements. In addition, the performance of the proposed platform will be compared with other commercial and open-source solutions to identify potential advantages and limitations.

**Author Contributions:** G.G.-G. performed conceptualization, methodology, software, validation, formal analysis, investigation, writing original draft, writing-review & editing, visualization, supervision. R.D. performed conceptualization, funding acquisition, methodology, project administration, supervision, validation, writing original draft. J.C. performed conceptualization, software, validation, formal analysis, investigation, writing original draft. I.G.-M. performed conceptualization, software, validation, formal analysis, investigation, writing original draft. All authors have read and agreed to the published version of the manuscript.

**Funding:** This research was funded by the European Commission grant number 740610 through the STOP-IT project (Strategic, Tactical and Operational Protection of critical water Infrastructures against physical and cyber Threats).

**Acknowledgments:** A special thanks to the Mekorot team for their collaboration and insights in the testing and validation of the tools composing the proposed platform.

**Conflicts of Interest:** The authors declare no conflict of interest.

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
