# Peer review of "CLAP: A Cross-Layer Analytic Platform for the Correlation of Cyber and Physical Security Events Affecting Water Critical Infrastructures"

_jcp, doi:10.3390/jcp1020020_

Round 1

Reviewer 1 Report

The manuscript describes a novel platform to address the problem of correlating cyber and physical security data. The proposed platform is composed of a series of sensors that collect input data from cyber and physical devices and correlate them based on security rules. A severity value from 1 to 10 is assigned to each correlated alarm with the aim of helping security analysts to define priorities and strategies to protect their infrastructures from cyber-physical threats. Overall , the manuscript is very well written and well detailed, the platform is presented by modules and explained accordingly. In addition, a series of attack scenarios are discussed to highlight the usage of the platform. There are however, some main aspects that could be improved in the paper:

  • Authors claim the platform is validated in Section 5.4 but there are no details on the stakeholders that validated the platform and the concrete results obtained. It is important to include this information at the end of this section either as a summary table or as a new section describing them.
  • Related works present some references that are quite all, it is advised to either keep  only updated ones. In addition, it is recommended to highlight the novelties of the proposed platform with regards to the state-of-the-art.
  • Some of the tools from the proposed platform are never cited. Please add references to all of them (whenever possible).

  • Some new solutions should be reviewed,  for example :

On the design of secure primitives for real world applications N Sklavos, N Kaaniche Microprocessors and Microsystems 80, 103614

Reviewer 2 Report

The paper presents CLAP, a Cross-Layer Analytic Platform developed for the correlation of cyber and physical security events affecting water CIs. The platform is used to detect cyber and physical attacks against a target infrastructure emulating a water CI with PLCs, and various security tools (IDS, SIEM, etc). The platform is composed of (i) a central node including a SIEM tool and a Real-time anomaly detector tool that receive cyber and physical data from security sensors; (ii) a physical detection module and (iii) a cyber detection module.

The related work section presents several works, however it could be enriched with some recent advances in the field of cyber physical attacks with respect to critical systems, see for example:

  • "Assessing IoT enabled cyber-physical attack paths against critical systems", Computers and Security, 2021, https://doi.org/10.1016/j.cose.2021.102316

  • "Autonomous mitigation of cyber risks in the Cyber–Physical Systems", Future Generation Computer Systems, 2021 https://doi.org/10.1016/j.future.2020.09.002

In addition, I would expect a more critical review / comparison of the proposed system with respect to the existing related work. For example, a table that summarizes the related works and briefly describes the differences and/or pros and cons of existing systems including the proposed system.

Concerning section 5.4 Platform Validation, I am not sure what is the actual contribution of this section. You describe some validation criteria and the methodology, but I do not see the results of the validation against those criteria. I would suggest to either add the validation results or reduce this section completely.

In a similar way, section 6.5 Preliminary Results is not what it states. It is rather a description of what you want to improve and not the actual results. What I would expect to see here are actual validation results showing for example the accuracy of the detection, success/failure to detect threats, execution time etc. Although examples are described in the previous subsections, there are no actual evaluation results regarding the effectiveness and the efficiency of the presented tool.

The table in Figure 4 is very hard to read.

Round 2

Reviewer 2 Report

All the comments have been sufficiently addressed in the revised version. I would only suggest a final proof reading.